# An Electromagnetic Lock Actuated by a Mobile Phone Equipped with a Self-Made Laser Pointer

**Jau-Woei Perng [1] and Tung-Li Hsieh [1,2,*]** 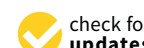

[1] Department of Mechanical and Electromechanical Engineering, National Sun Yat-sen University, Kaohsiung 80424, Taiwan; jwperng@faculty.nsysu.edu.tw

[2] General Education Center of Wenzao Ursuline University of Languages, Wenzao Ursuline University, Kaohsiung 80793, Taiwan

* Correspondence: tunglihsieh@gmail.com; Tel.: +886-7-342-6031

**Abstract:** The main purpose of this study was to create an acousto-optic control lock device to convert electrical signals with a specific sound command using an acousto-optic conversion module, thereby improving the reliability and safety of opening or closing remote controlled door locks, such as car central locks or rolling doors. We used music playing through a smart phone speaker to create a special laser pointer to connect with the smart phone's auxiliary input. The laser pointer (wavelength of 630–650 nm and maximum output of 5 mw) lights up when the smart phone's music starts playing at a music frequency matching the light frequency. When the solar panel receives light, it converts the frequency of the light signal into an electrical frequency signal. The current is amplified using the power amplifier and then the amplified current flows to the sound recognition module. The sound recognition module performs audio comparison on the set sound signal, and once the comparison is correct, the output voltage activates the electromagnetic switch on the door to open or close it.

**Keywords:** laser pointer; electromagnetic lock; sound recognition module

---

## 1. Introduction

Traditional electronic remote controls [1,2] have been widely used in iron rolling doors, car central locks, scooter electronic locks, and various types of door locks. The remote control is used to control the lock. A signal transmitter is configured in the remote control to send the control signal to open or close the door to a signal receiver. The signal receiver is usually connected to an actuation circuit, which then turns the connection mechanism on or off to open or close the door. Most remote controls use an infrared signal or a radio frequency (RF) signal [3–5] as the control signal to open or close the door. However, this requires the user to carry an additional device, which can be lost, stolen, or damaged easily. Losing or forgetting the remote control would result in being locked out and the electronic remote controls fail when their battery power is depleted. With technological advancement, electronic remote controls are susceptible to replication and theft, and their security is gradually being challenged. Such incidences cause inconvenience to the users of traditional remote controls.

Traditional electronic remote controls need to be improved to eliminate the inconvenience and lack of security. To overcome these issues, in this study we formulated a method so that users do not need to carry an electronic remote control with them at all.

## 2. Materials and Methods

The proposed model is based on a control lock device: an audio-coupled, laser-actuated, electromagnetic lock device mounted on a door plank used to open or close the door using audio. The audio-coupled, laser-actuated, electromagnetic lock device consists of a transmitting unit, a receiving

unit, a recognition unit, and an actuation unit. The transmitting unit of the proposed device is composed of an acoustic receiver and an acousto-optic conversion module, and the receiving unit is configured with an optical receiver. The acousto-optic conversion module converts a specific audio command signal into an electrical signal, thereby improving the reliability and safety of door locking or unlocking. The transmitting unit can be embedded in a mobile phone to open or close the door lock. Thus, users would not need to carry the remote control, increasing convenience of use.

### 2.1. Apparatus

The energy conversion module contains a power amplifying circuit, signal input and output terminals, and a power terminal. When the energy conversion module is operating, its temperature increases. The positive and negative poles of a solar panel are connected to the input pins of the energy conversion module using wires. Because ambient light in the environment can affect a solar panel and generate noise, the panel was modified to include a masking structure. The energy conversion module adopts a low-voltage LM386 chip (U.S. National Semiconductor, city, state abbrv. if USA, country) [6,7], which is equipped with a pin function (Figure 1). These chips are commonly used in low-voltage consumer products to minimize peripheral chip components. The LM386 chip's voltage gain is typically set to 20. However, by adding an external resistor and a capacitor between pins 1 and 8, the voltage gain can be arbitrarily adjusted up to a maximum value of 200 (Figure 2).

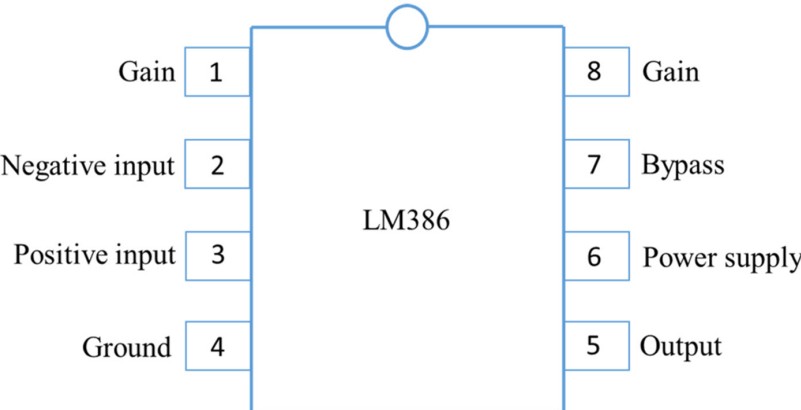

**Figure 1.** LM386 chip function pin, which features low static power consumption (approximately 4 mA) that is suitable for battery power supply, a wide operating voltage range (4–12 V or 5–18 V), fewer peripheral components, an adjustable voltage gain of 20–200 V, and a low distortion rate.

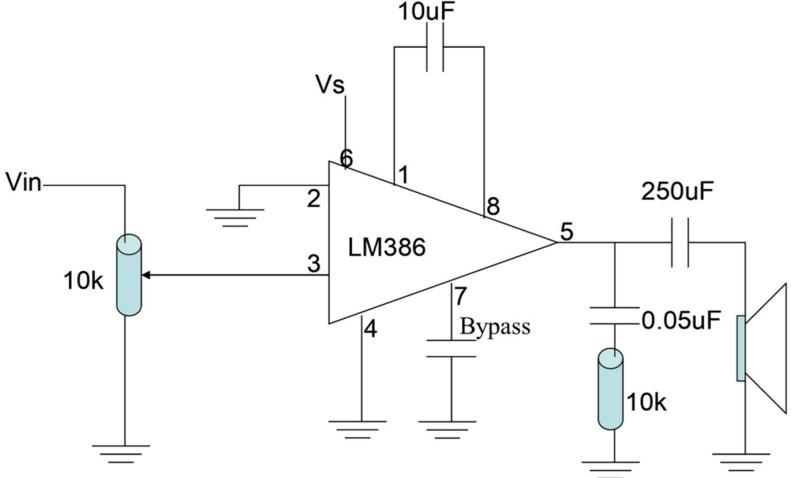

**Figure 2.** Circuit diagram with a gain of 200 V.

### 2.2. Electromagnetic Lock Mechanism Modeling

Figure 3a presents the mathematical model [8] with physical parameters, with the equation of motion as follows:

$$m\ddot{x} + c\dot{x} + kx = f(t) \tag{1}$$

where $m$, $c$, and $k$ represent the mass, damping coefficient, and spring constant, respectively, which are the physical quantities of the system's mass, damping, and spring components, respectively. Figure 3b presents a mathematical model with modal parameters. By dividing Equation (1) by $m$ and substituting the variables, the equation of motion for the physical parameters can be rewritten in modal parameter form as follows:

$$\ddot{q} + 2\xi\omega_n\dot{q} + \omega_n^2 q = N(t) \tag{2}$$

where

$$\omega_n = \sqrt{\frac{k}{m}}, \xi = \frac{c}{c_c}, c_c = 2m\omega_n = 2\sqrt{mk}\, q(t) = x(t), \text{ and } N(t) = \frac{f(t)}{m},$$

where $c_c$ is the critical damping coefficient, which denotes the damping ratio, $\omega_n$ denotes the natural frequency, $x(t)$ is the physical coordinate, $f(t)$ defines the physical force, and $q(t)$ is the modal coordinate.

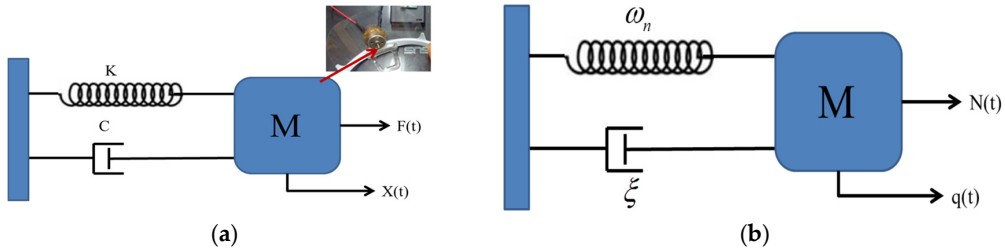

(a)          (b)

**Figure 3.** Mathematical model with (**a**) physical parameters and (**b**) with modal parameters.

Each conversion between domains, such as the conversion from a mathematical model with physical parameters to one with modal parameters, or the conversion of a mathematical model with modal parameters into one with frequency parameters, has a corresponding equation (Figure 4).

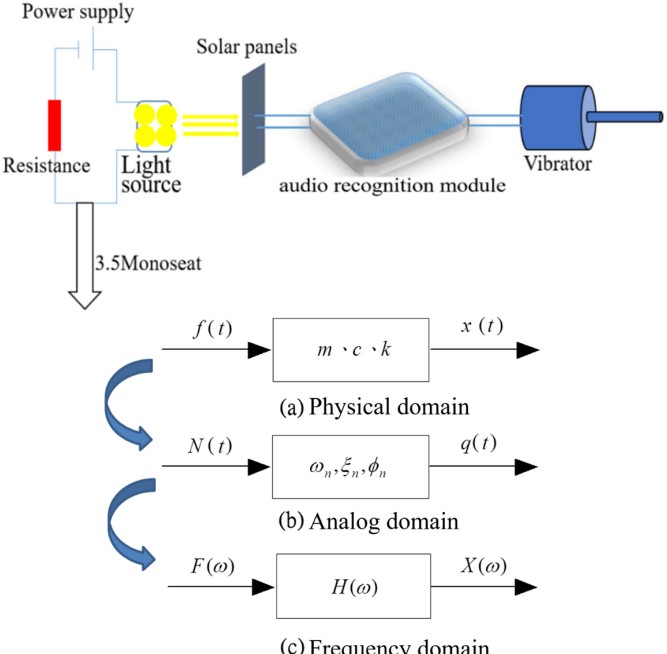

**Figure 4.** Circuit diagram of an audio-coupled, laser-actuated, electromagnetic lock device.

*2.3. Response Analysis*

Harmonic excitation force was set as the external force, $f(t) = Fe^{i\omega t}$. By substituting $x(t) = Xe^{i\omega t}$ into the equation of motion, the frequency response function is obtained:

$$H(\omega) = \frac{X}{F} = \frac{1}{(k - m\omega^2) + i(\omega c)} = \frac{1/m}{(\omega_n^2 - \omega^2) + i(2\xi\omega_n\omega)} \tag{3}$$

Using Equation (3), the frequency response function can be converted into a function of physical or modal parameters, which is related to the harmonic excitation frequency $\omega$.

For the transient response analysis, if the input condition $f(t)$, initial conditions $x_0$ and $v_0$, and system content (e.g., the physical and modal parameters) are known, the time-domain output response of the system can be obtained using the following equation:

$$\begin{aligned} x(t) &= e^{-\xi\omega_n t}(A\cos\omega_d t + B\sin\omega_d t) + \int_0^t f(t)h(t-\tau)d\tau \\ \omega_d &= \omega_n\sqrt{1-\xi^2}; h(t) = \frac{1}{m\omega_d}e^{-\xi\omega_n t}\sin\omega_d t \end{aligned} \tag{4}$$

where $\omega_d$ denotes the damped natural frequency; A and B are random constants defined by the initial condition; and $h(t)$ represents the unit impulse response function of the system.

For the spectrum response analysis, if the frequency domain can be presented as the power spectral density (PSD) [9] function $x(t)$ and the frequency response function is known, the frequency domain response can be obtained through spectrum response analysis:

$$G_{xx}(\omega) = |H(\omega)|^2 G_{ff}(\omega) \tag{5}$$

where $G_{xx}(\omega)$ is the physical coordinate and $x(t)$ denotes the PSD function.

Figure 4 depicts the circuit of an audio-coupled, laser-actuated, electromagnetic lock device. In the creation of the proposed method, the audio-coupled, laser-actuated, electromagnetic lock included a light-emitting diode (LED) and a modulation circuit. The LED was used to generate an optical signal, and the modulation circuit was electrically connected to the audio receiver and LED. Next, the receiving unit, which included an amplification circuit, was electrically connected to an optical receiver to amplify its electrical output signal. In the third implementation, the optical receiver, namely a solar panel, was included. In the fourth implementation, the solar panel and door lock were mounted on a door plank. Then, the recognition unit was configured with a comparator to compare the electrical signal with a preset signal, which generated an actuation signal if the comparison result was the same. Next, the recognition unit, which includes storage, stores the preset signal. In the seventh implementation, the actuation unit was created, which included an electromagnetic actuator for opening or closing the door lock. In the eighth implementation, the transmitting unit was built into a portable electronic product, either a smartphone or a tablet. In the ninth implementation, the audio receiver, a monophonic plug, was inserted in the sound port of a portable electronic product.

As described earlier, the acousto-optic control lock device of the proposed model can open or close the door lock using a specific audio command signal, thereby improving the reliability and safety of the door lock, as well as preventing a lock key from being replicated or stolen. The transmitting unit embedded in the mobile phone enables the lock to be opened or closed without the use of a remote control, increasing convenience of use for the user. The proposed model is suitable for long-distance transmission because an audio command signal is replaced with an optical signal and a specific audio signal is transmitted, thereby increasing convenience.

The objective of the modal system analysis and testing was to develop specific analysis procedures and testing and measurement methods to obtain a mathematical model of the sound vibration mechanism. This model can be presented in the form of a mathematical model with modal or physical parameters. The analysis process flowchart is presented in Figure 5.

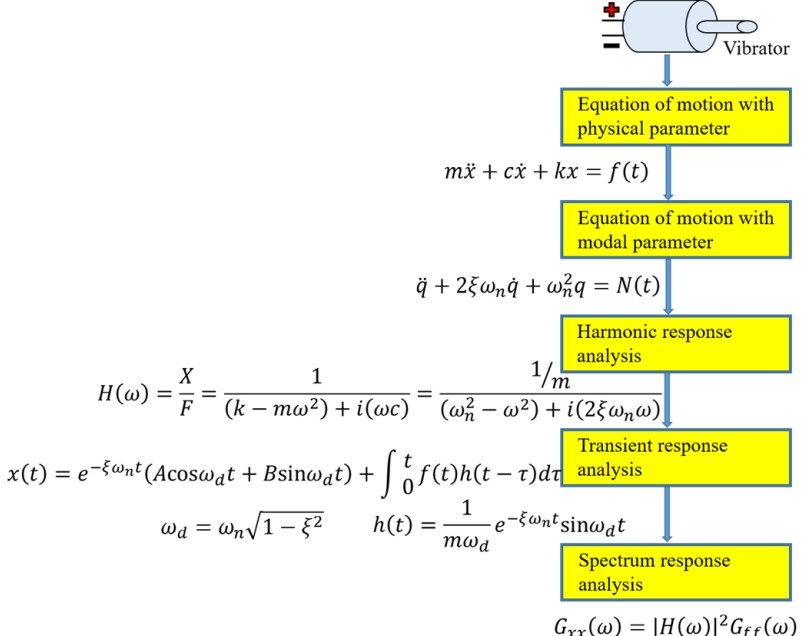

**Figure 5.** Flowchart of analysis of the sound vibration mechanism.

## 3. Results

### 3.1. Speech Endpoint Detection

We adopted a single channel in this experiment. Because different volumes generate different amplitudes, audio signals are normalized in order to unify amplitudes and obtain a valid frame range for subsequent signal processing. Human speech often contains aspiration or friction, which can complicate the detection of sound energy due to its low energy. The zero crossing rate in the endpoint detection method [10] can be employed to correct the endpoint range and extract an entire syllable (Figure 6).

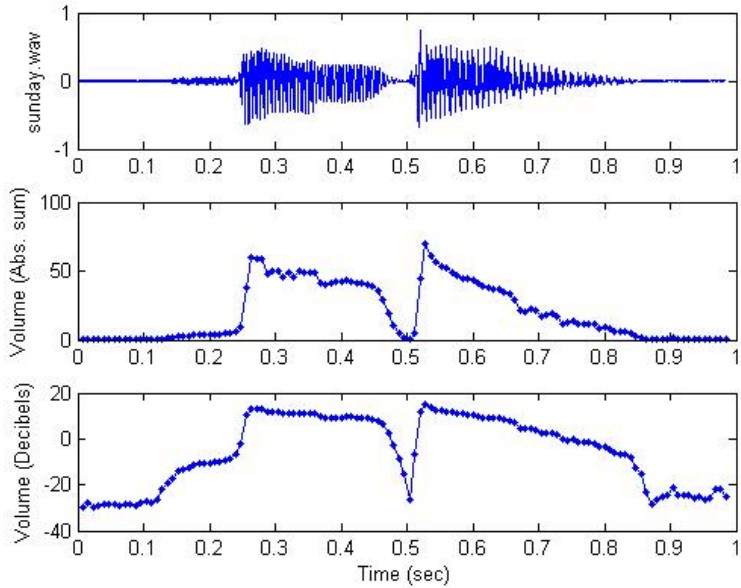

**Figure 6.** Sampled voice signals in the time domain after ZCR calculation.

*3.2. Feature Extraction*

Signal characteristics are difficult to recognize by observing changes in the amplitude of an audio signal over time (Figure 7). Converting an audio signal into a spectrogram (Figure 8) allows sonic characteristics to be identified [11]. The amplitude value of an audio signal in the time domain is often converted into an energy distribution in the frequency domain for observation. Various energy distributions in the frequency domain represent different speech characteristics. A signal changes rapidly and constantly over time, leading to inaccurate observation. Therefore, the most commonly used technique is conversion of the audio signal from the time domain to the frequency domain, enabling the identification of the spectral characteristics of various sounds through their energy distribution. The spectrum is a representation of a time domain signal in the frequency domain and can be obtained by performing FFT on the signal [12–17]. The result is presented as a spectrogram, with the amplitude or phase as the vertical axis and the frequency as the horizontal axis (Figure 9).

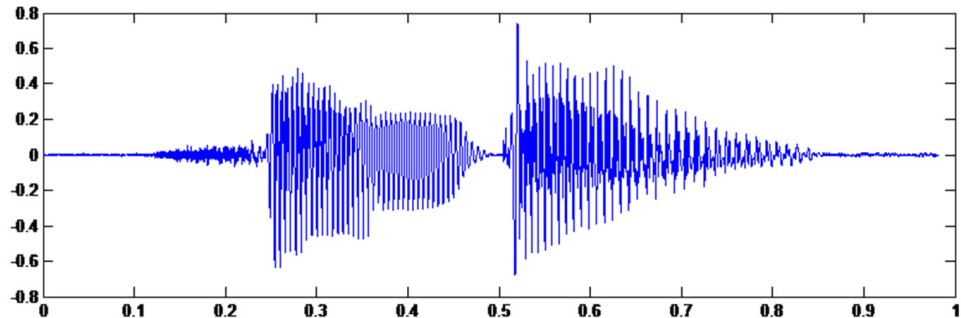

**Figure 7.** Amplitude value of audio signal, where *fs* (44,100) is the sampled frequency.

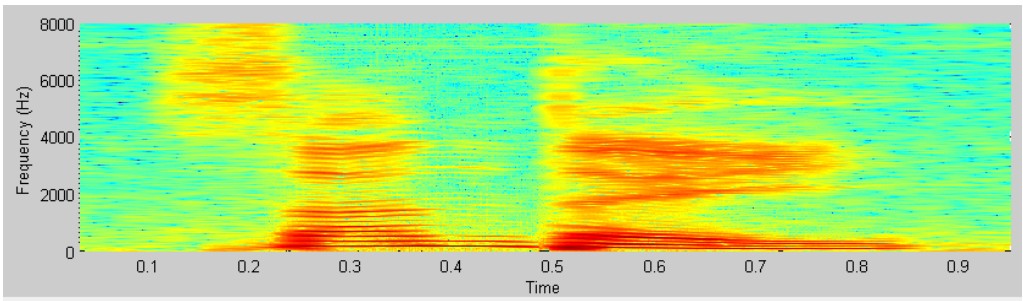

**Figure 8.** Audio signal spectrogram.

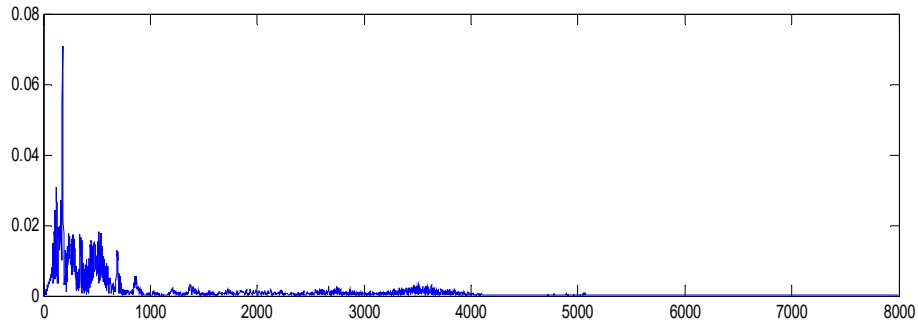

**Figure 9.** Audio signal spectrum.

*3.3. Filter Design*

Figure 10 shows the design of a Butterworth filter [18–22] with a cutoff frequency of 1000 Hz. The plot depicts the magnitude frequency response of the filter.

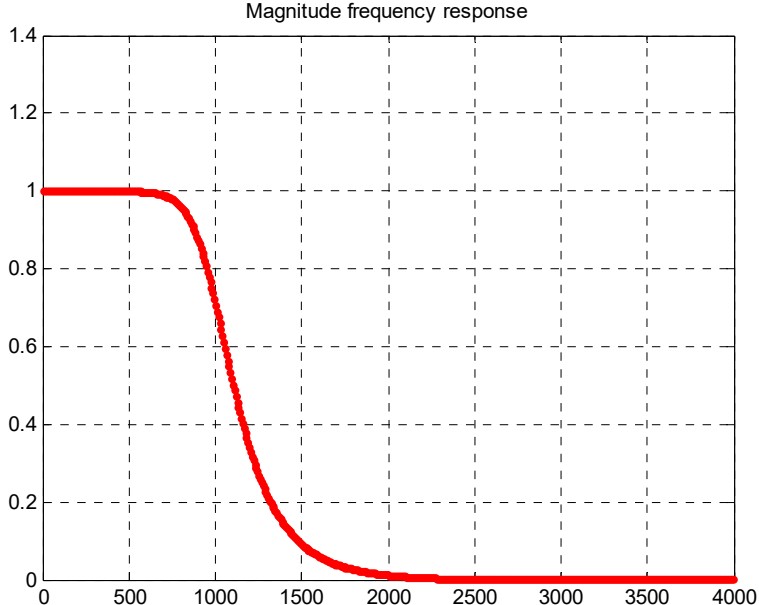

**Figure 10.** Magnitude frequency response of the filter.

When the order of the filter is larger, the filtering effect is better at the cost of longer computation time. A lower order leads to a shorter computation time and a less desirable filtering effect. Figure 11 demonstrates the magnitude frequency response as a function of the order of the Butterworth filter. Figure 11 shows that when the order increases from one to eight, the magnitude frequency response sharpens at the cutoff frequency of 1000 Hz.

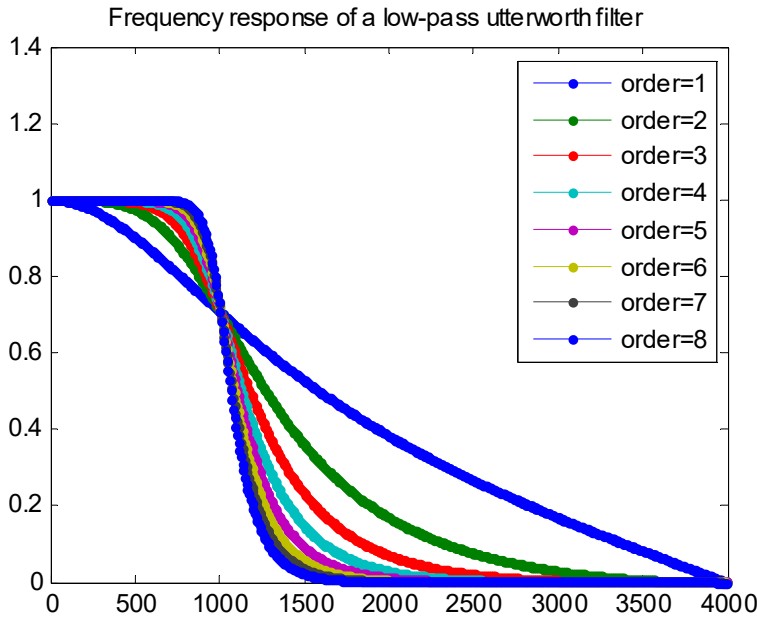

**Figure 11.** Frequency response of a low-pass Butterworth filter.

The original audio was passed through a low-pass filter with a cutoff frequency of 1000 Hz to test whether it can filter the treble (Figure 12). Figure 13 depicts the difference between the original signal and the filter output signal, indicating that the treble is almost removed. So, we can apply the audio to see if the high-frequency components can be removed.

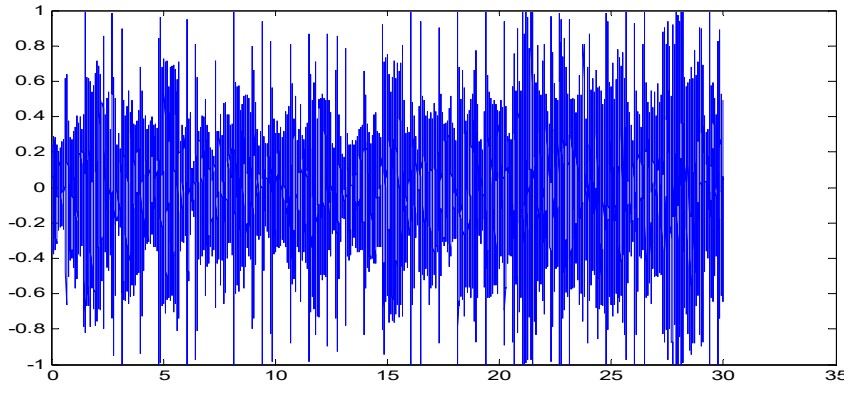

**Figure 12.** Original audio.

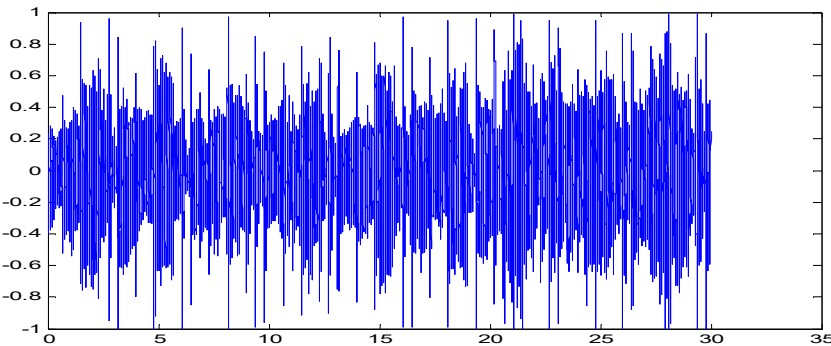

**Figure 13.** A low-pass filter with a cutoff frequency of 1000 Hz.

If we set the cutoff frequency to 100 Hz, then the output signal is almost inaudible unless we use a speaker with a subwoofer, as shown in Figure 14. After applying the low-pass filter at a cutoff frequency of 100 Hz, most of the sounds were removed, except the bass.

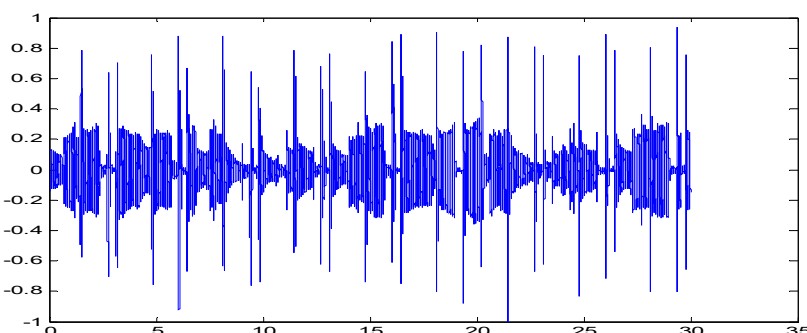

**Figure 14.** A low-pass filter with a cutoff frequency of 100 Hz.

The low-pass chopper functions to pass the low-frequency signal and attenuate the high-frequency signal, which is suitable for high-frequency noise. For example, if the temperature or flow sensor has a low frequency, the low-pass chopper can be used to remove electrical noise generated by a motor [23–26].

A digital filter can be represented by two parameter vectors, a and b, where the lengths of a and b are p and q, respectively, and the first element of a is always 1, as follows:

$$a = \left[1, a_2, \ldots a_p\right]$$

$$b = \left[b_1, b_2, \ldots b_q\right]$$

If we apply a digital filter with parameters a and b to a stream of discrete-time signal x[n], the output y[n] should satisfy the following equation:

$$y[n] + a_2y[n-1] + a_3y[n-2] + \ldots + a_px[n-p+1] = b_1x[n] + b_2x[n-1] + \ldots + b_qx[n-q+1]$$

Or equivalently, we can express y[n] explicitly as:

$$y[n] = b_1x[n] + b_2x[n-1] + \ldots + b_qx[n-q+1] - a_2y[n-1] - a_3y[n-2] - \ldots - a_px[n-p+1]$$

The preceding equation is somewhat complicated. We provide some more specific examples to facilitate understanding.

First, if we have a filter with the parameters a = [1] and b = [1/5, 1/5, 1/5, 1/5, 1/5], then the output of the filter is y[n] = (x[n] + x[n − 1] + x[n − 2] + x[n − 3] + x[n − 4])/5.

Figure 15 shows the different levels of filtering effects at different cutoff frequencies. The treble part is almost deleted after filtering. Only the sound of the bass survives, and then the bass sound that appears through these regular rules allows us to track the beat.

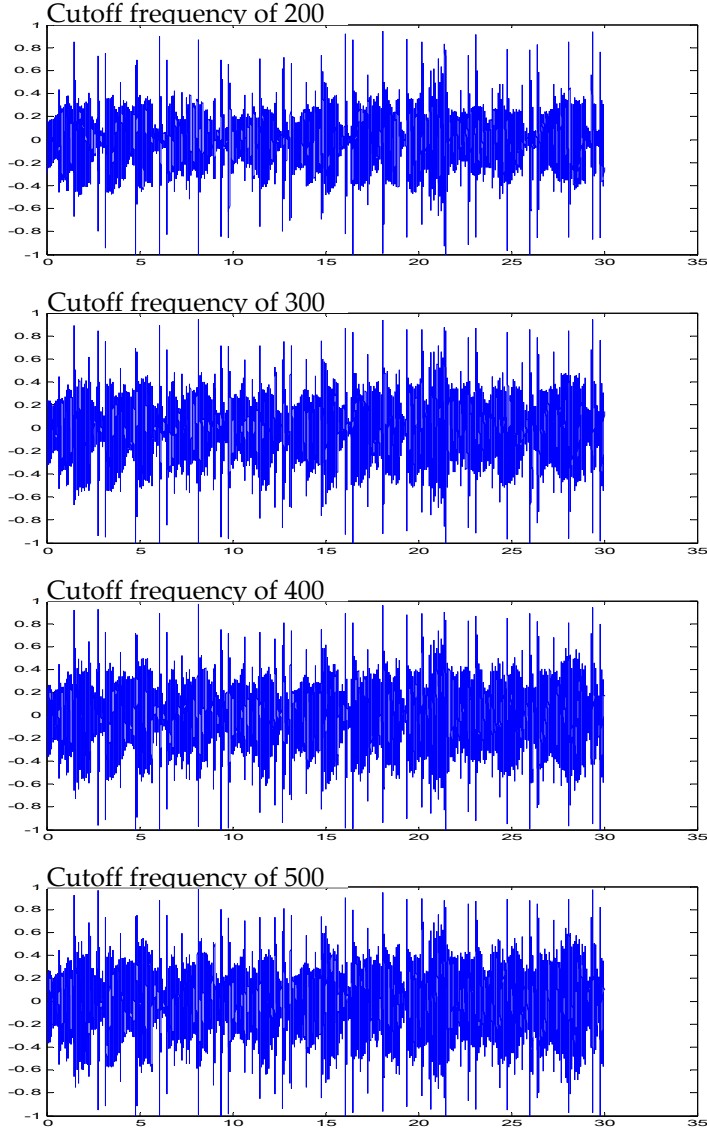

**Figure 15.** The different levels of filtering effects at different cutoff frequencies.

The result is a simple filter set y[n] as the average of the preceding five points of the input signals. In fact, this is a low-pass filter [27–31], since after the averaging operator, the high-frequency component is averaged out, while the low-frequency component is retained. The effect of a low-pass filter is like putting a paper cup over one's mouth while speaking, generating a murmuring-like, indistinct sound.

## 4. Discussion

The acousto-optic control lock (Figure 16) in the proposed device can open or close a door lock using a specific audio command signal, thereby improving the reliability and safety of the door lock and preventing a key from being replicated or stolen. The transmitting unit, embedded in a mobile phone, enables a door lock to be opened or closed without a remote control, which is convenient for the user. This model is suitable for long-distance transmission because the audio command signal is replaced with an optical signal and a specific audio signal is transmitted, thereby further enhancing convenience. The benefits of the proposed model are as follows: (1) elimination of inconvenience caused when keys are forgotten; (2) elimination of the need to carry multiple keys due to the laser-actuated electromagnetic lock device being coupled with mobile audio; (3) avoidance of the need to open a door with a key, as with a mobile phone a user can open multiple doors and set different opening passwords for each door; and (4) a significant reduction in the need for metallic materials for creating keys and elimination of the requirement for electroplating, which damages the environment, thus mitigating environmental pollution. The performance factors of an audio-coupled, laser-actuated, electromagnetic lock device are as follows: (1) An audio file is equivalent to a key. Users can easily set up or change different audio files on their own as the key. It is even more convenient than changing keys for traditional door locks. (2) The device allows for high confidentiality. As the audio file in the mobile phone is outputted through the laser carrier waves emitted by the laser pointer at the same audio frequency, only laser light is emitted when the phone plays the audio file, whereas no sound is generated. Hence, external remote sensors cannot detect the audio information. (3) Compared to traditional door locks, this device enables users to carry fewer bulky keys with them. (4) For traditional door locks, once the key is lost, both the lock and key have to be replaced. However, the photoelectric lock and laser pointer of this device are independent of one another and need not be replaced simultaneously. (5) The effective distance of the laser pointer (wavelength of 630–650 nm and maximum output of 5 mw) is <100 m and the time latency is about 30 ms. Table 1 shows the performance comparison of various locks

**Table 1.** The performance comparison of various locks.

| | Traditional Door Locks | Radio Frequency (RF) Electronic Remote Lock | Optical Electromagnetic Lock |
|---|---|---|---|
| Unlock mode | key | RF signal | laser pointer |
| Remote open | X | V (10 m) | V (100 m) |
| safety | Easy to destroy and copy | RF is susceptible to strong wave disturbances Easy to copy and steal | Laser is not easily interfered with or copied |
| Convenience | Door locks or keys are damaged and must be replaced together | It is not easy to copy the program after losing it | Audio can be easily copied |
| Security | Easy copying from mobile audio | Easy to destroy from the outside | Because the electromagnetic lock is in the door, it is difficult to break |
| Interactivity | X | V | V |
| cost | NT $100–$200 | NT $3000–$4000 | NT $1000 or less |

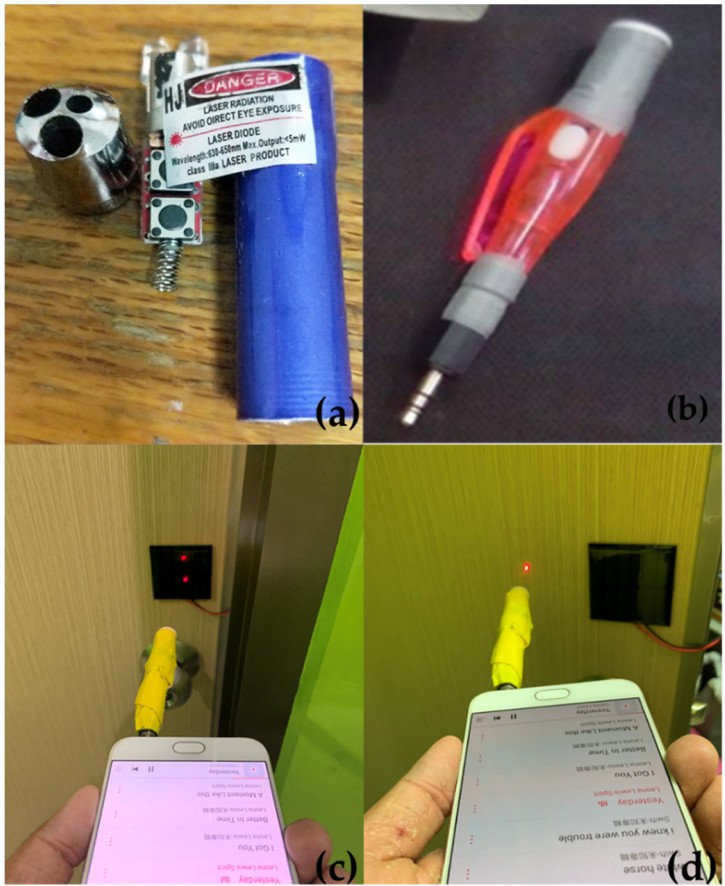

**Figure 16.** The composition and application of acousto-optic control locks. (**a**) Disassembly of a laser diode. (**b**) The circuit combination of a 3.5 monoseat and laser diode. (**c**) Laser pointer combined with mobile phone audio for unlocking. (**d**) The door lock is opened.

**Author Contributions:** T.-L.H. and J.-W.P. conceived of the presented idea. T.-L.H. developed the theory and performed the computations. T.-L.H. verified the analytical methods. J.-W.P. encouraged investigation and supervised the work. All authors discussed the results and contributed to the final manuscript.

**Funding:** This research received no external funding.

**Acknowledgments:** This research has been reviewed by the IEEE ECICE 2019 committee, where it was recommended to be submitted to *Electronics*.

**Conflicts of Interest:** The authors declare no conflict of interest.

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
