# Peer review of "An Electromagnetic Lock Actuated by a Mobile Phone Equipped with a Self-Made Laser Pointer"

_electronics, doi:10.3390/electronics8121524_

Round 1
Reviewer 1 Report
This research proposed an electromagnetic lock utilizing mobile phone audio and laser light. Different from the current remote control using cloud computing, the proposed lock more like a variant of near field wireless sensing appliance. The proposed lock has several advantages rather than existing work with RF signal, such as multi-functions, elimination of carrying extra keys, and reduction of materials.
The paper is well represented, while I have some small concerns:
Security. Please discuss more about the security of the approach, since it is the goal of a lock. For example, how to deal with the monitor that might record the audio data remotely? With the copy of the audio, can the hacker open the lock? It is better to show the performance of the lock to verify the feasibility, such as: time latency and effective distance.
Author Response
請參閱附件。

Reviewer 2 Report
The paper describes an electromagnetic lock activated by a mobile phone and a built-in laser pointer.
I would suggest the authors to rephrase the paper’s title (if is possible at this stage) to make it more comprehensible. If I may make a suggestion, a title such as “Electromagnetic lock actuated by a mobile phone equipped with a self-made laser pointer” might better express what the authors describe. In any case, it is up to the authors and the editor to adopt (or not) this suggestion.
Moderate changes regarding the use of English are necessary. In any case, the first four sentences of the abstract (lines 11-14) have to be rephrased. The same applies to the first two sentences of the introduction (lines 25-27). Moreover, a “not” seems to be missing in the sentence in lines 31-32.
Sections 2.2 and 2.3 should become sorter since parts of those sections are rather trivial. Eqs. (1) to (5) should be given with very short comments and figs. 4 and 5 could be combined to one. In any case, the authors should relate the analysis with the operation of the proposed device
I understand that lines 123 to 136 describe various implementations of the proposed device. I think that the word “implementation” fits better than the word “embodiment”. The authors could also describe one or more implementations in more detail.
There is no reference to the characteristics of the laser pointer (e.g. emitted wavelength / power)
Section 3.3 should only include analysis and facts directly related to the design and the parameters of the proposed device. For example, lines 198 to 211 and 231 to 247 are parts of basic filter theory and can be omitted.
Though I basically agree with the advantages included in section 4 (“Discussion”), the fact is that mobile phones are, on the other hand, battery-dependent and more subject to theft than electronic keys. Including those comments would make the comparison to conventional remote controls more fair.
Reviewer 3 Report
Please include comparisons in your paper to demonstrate the proposed method shows better performance to conventional methods.
Round 2
